# Unraveling the Mechanism of Platelet Aggregation Suppression by Thioterpenoids: Molecular Docking and In Vivo Antiaggregant Activity

**DOI:** 10.3390/biomimetics8080570

**Published:** 2023-11-27

**Authors:** Liliya E. Nikitina, Pavel S. Bocharov, Alexander A. Ksenofontov, Elena V. Antina, Ilmir R. Gilfanov, Roman S. Pavelyev, Olga V. Ostolopovskaya, Inna V. Fedyunina, Zulfiya R. Azizova, Svetlana V. Pestova, Evgeniy S. Izmest’ev, Svetlana A. Rubtsova, Sergei V. Boichuk, Aigul R. Galembikova, Elena A. Smolyarchuk, Ilshat G. Mustafin, Airat R. Kayumov, Aleksandr V. Samorodov

**Affiliations:** 1General and Organic Chemistry Department, Kazan State Medical University, 420012 Kazan, Russia; olga-ov.kirill@mail.ru (O.V.O.); inovit@mail.ru (I.V.F.); gzr27@yandex.ru (Z.R.A.); 2Institute of Fundamental Medicine and Biology, Kazan Federal University, 420008 Kazan, Russia; ilmir.gilfanov@gmail.com (I.R.G.); rpavelyev@gmail.com (R.S.P.); boichuksergei@mail.ru (S.V.B.);; 3G.A. Krestov Institute of Solution Chemistry of the Russian Academy of Sciences, 153045 Ivanovo, Russia; bochpavl@gmail.com (P.S.B.); ivalex.09@mail.ru (A.A.K.); eva@isc-ras.ru (E.V.A.); 4Department of Inorganic Chemistry, Ivanovo State University of Chemistry and Technology, 153045 Ivanovo, Russia; 5Varnishes and Paints Coatings Department, Kazan National Research Technological University, 420015 Kazan, Russia; 6Medical Chemistry Laboratory, Institute of Chemistry, Komi Scientific Centre, Ural Branch of Russian Academy of Sciences, 167000 Syktyvkar, Russia; pestova-svetlana89@mail.ru (S.V.P.); evgeniyizmestev@rambler.ru (E.S.I.); rubtsova-sa@chemi.komisc.ru (S.A.R.); 7Department of Pathology, Kazan State Medical University, 420012 Kazan, Russia; ailuk000@mail.ru; 8Department of Radiotherapy and Radiology, Russian Medical Academy of Continuous Professional Education, 125993 Moscow, Russia; 9Department of Pharmacology, Sechenov First Moscow State Medical University (Sechenov University), 125993 Moscow, Russia; makval81@rambler.ru; 10Biochemistry Department, Kazan State Medical University, 420012 Kazan, Russia; ilshat64@mail.ru; 11Department of Pharmacology, Bashkir State Medical University, 450008 Ufa, Russia; avsamorodov@gmail.com

**Keywords:** *S*-containing monoterpenoids, molecular docking, integrin α_IIb_β_3_, antiaggregant activity

## Abstract

Natural monoterpenes and their derivatives are widely considered the effective ingredients for the design and production of novel biologically active compounds. In this study, by using the molecular docking technique, we examined the effects of two series of “sulfide-sulfoxide-sulfone” thioterpenoids containing different (e.g., bornane and pinane) monoterpene skeletons on the platelet’s aggregation. Our data revealed that all the synthesized compounds exhibit inhibitory activities on platelet aggregation. For example, compound **1** effectively inhibited platelet activation and demonstrated direct binding with CD61 integrin, a well-known platelet GPIIb-IIIa receptor on platelets. We further examined the antiaggregant activity of the most active compound, **1**, in vivo and compared its activity with that of acetylsalicylic acid and an oral GPIIb-IIIa blocker, orbofiban. We found that compound **1** demonstrates antiaggregant activity in rats when administered per os and its activity was comparable with that of acetylsalicylic acid and orbofiban. Moreover, similarly, tirofiban, a well-known GPIIb-IIIa blocker, compound **1**, effectively decreased the expression of P-selectin to the values similar to those of the intact platelets. Collectively, here, we show, for the first time, the potent antiaggregant activity of compound **1** both in vitro and in vivo due to its ability to bind with the GPIIb-IIIa receptor on platelets.

## 1. Introduction

Abnormal changes in blood coagulation leading to hemostasis disorders are the key elements of various obstetric and surgical pathologies, and cardiovascular, cerebrovascular, infectious, and immune diseases [1,2,3,4]. Nevertheless, the prevention and treatment of thrombosis and hemorrhagic conditions remain challenging and require a deeper insight into the molecular mechanisms of blood coagulation and its regulation. Recently, significant results have been obtained in studies clarifying vascular platelet hemostasis the coagulation factors’ molecular mechanism conversions in plasma hemostasis. It is worthwhile noting that most of the drugs suppressing the activity of platelets available to date do not guarantee the effective prevention or treatment of thrombosis. Antiplatelet agents as medications suppressing the activity of platelets are used to prevent thrombotic events. Aspirin is the most common and widely used antiplatelet agent. Thus, the resistance to aspirin of up to 61% of patients has been reported; it is the most common and widely used anti-aggregation agent, and acts as an irreversible blocker of the cyclooxygenase enzymes and thromboxane A2 synthesis inhibitor [5]. Resistance to clopidogrel, an inhibitor of well-known P2Y_12_ platelet receptors, was reported to be in the range 5–45% [6]. Therefore, the development of new agents for corrections of hemostasis system disorders is strictly required.

Obviously, agents affecting both the activity of cell receptors and the thrombogenic properties of the cell’s phospholipid surface may become very promising compounds. Terpenes are a vast class of substances found in all living organisms. Previous studies described the membrane-stabilizing properties of terpenes. Their molecules are rigid, ambiphilic and provide Van der Waals interactions with phospholipids of cell membranes, leading to the stabilization of the cell membrane and the effects on its thrombogenic properties’ manifestation. In our previous paper, the interactions of thioterpenoids with phospholipid membranes were studied using various NMR techniques. The findings of this study indicated that bicyclic thioterpenoids have a membrane location, which is shifted somewhat in the direction of the lipid–water interface. Such a location may shield the compound from interactions with hydrophobic lipid segments. Detailed NMR studies on dodecyl phosphocholine (DPC) as the membrane model revealed that *S*-containing monoterpenoids bind to DPC micelles’ surface. This binding reinforces the mechanical properties of the cell membranes and prevents destabilization and subsequent clot formation on the phospholipid surface. Moreover, to examine the detailed atomistic picture of thioterpenoid–SDS micelle interactions, classical molecular dynamics simulations were performed [7]. A visual examination of the pinnae sulfoxide–micelle complex indicated that sulfoxide is embedded with its bicyclic fragment inside the SDS micelle, whereas the hydrophilic –SO(CH2)2OH fragment of sulfoxide is located on the outer part of the SDS micelle and is in contact with the solvent. Most likely, for thioterpenoids, both mechanisms of action are realized—receptor and membrane factors

Recently, we have shown that sulfur-containing monoterpenoids, to a greater extent than oxygen and nitrogen-containing analogues, can reduce spontaneous and induced platelet aggregation [8]. Molecular docking indicated that the binding force with the platelet P2Y_12_ receptor depends strongly on the contained heteroatom in the structure of monoterpenoids, where it was found to be stronger in the presence of sulfur and weaker in the presence of oxygen and nitrogen atoms.

In our previous work, two series of “sulfide-sulfoxide-sulfone” thioterpenoids, differing from each other in the structure of their terpenic skeletons, have been described [9]. The obtained compounds, **1**–**3**, present bornane series compounds; meanwhile, compounds **4**–**6** present the pinane series (Figure 1).

Using molecular docking, the affinity of thioterpenoids **1**–**6** to the P2Y_12_ receptor was studied using their antioxidant activity, and the antiaggregational and anticoagulant activity of **1** and **4** sulfides as the most promising compounds of multifactorial action have been evaluated.

P2Y_12_ receptors are located in the platelet membrane and are an important regulator of platelet aggregation. Adenosine diphosphate (ADP) is a key element in the activation of purinergic receptor P2Y_12_. The P2Y12 receptor acts via a Gi-coupled pathway that inhibits adenylyl cyclase, and, as a result, lowers cyclic-adenosine monophosphate (AMP) levels. Cyclic-AMP lowering results in increased Ca^2+^ signal generation. In addition, P2Y_12_ stimulates a phosphatidylinositol 3-kinase (PI3-K) pathway, putatively involving the γ-isoform of PI3-K, which is known to be activated by βγ regulatory subunits released from Giα. The activity of PI3-K completes ADP-induced aggregation by enhancing the secretion reaction and activating integrin α_IIb_β_3_ [10]. Therefore, we investigated the potential of compounds **1**–**6** to act as inhibitors of platelet aggregation by evaluating their interaction with the P2Y_12_ receptor protein. Docking results demonstrated that the structural and energetic features of the interaction of compounds **1**–**6** with P2Y_12_ point towards their potential ability to act as inhibitors of platelet aggregation. Terpenoids **1**–**6** are retained in the P2Y_12_ pocket via van der Waals and hydrophobic interactions, as well as hydrogen bonds to the reactive groups of amino acid residues. Thus, the modification of functional groups and the structure of the terpene skeleton did not significantly affect the hydrophobicity, or, consequently, the affinity for the discovered P2Y_12_ pocket of compounds **1**–**6**. ChemScore d*G* analysis showed that all the structural features of terpenoids **1**–**6** did not significantly affect the binding energy to P2Y_12_.

Moreover, the studied sulfides, **1** and **4**, have shown high antioxidant activity as revealed via lipid peroxidation process inhibition in a non-cellular-substrate-containing animal lipids.

The in vitro data from the comparison of the effect of compounds **1** and **4** and commercial drugs on platelet aggregation induced by ADP showed that compound **1** exhibits antiaggregatory activity exceeding the values of that of acetylsalicylic acid by 1.9 times, and compound **4** exhibits antiaggregatory activity at the level of that of acetylsalicylic acid. At the same time, compounds **1** and **4** significantly shortened the latency period in comparison with that of the control and the comparison drug. Additionally, compounds **1** and **4** caused hypocoagulation, increasing APPT by 5.6% and 9.2%, compared with that of the control sample, and did not affect the concentration of fibrinogen and the prothrombin time.

Although there are many types of glycoproteins on the platelet surface, integrin α_IIb_β_3_ (also known as GPIIb/IIIa) is considered to be the most significant.

In this work, we have evaluated the affinity of thioterpenoids **1**–**6** to integrin α_IIb_β_3_ using molecular docking. Integrin is an important receptor responsible for many significant processes in the human body. Integrin determines the shape of the cell and its mobility, which are involved in the regulation of the cell cycle. In addition, part of the integrin is responsible for fibrinogen binding, which directly affects blood clotting, blood clot formation, etc. Integrin binding to various ligands can have a significant impact on its activity, which in turn will depend on the specific binding site. In intact platelets, α_IIb_β_3_ integrins are stored in a curved form. Integrin activation can be triggered in two ways—intracellularly and extracellularly. First of all, integrin activation is initiated by thrombin. Thrombin acts on protease-activated G-protein receptors (GPCR), which leads to an increase in the concentration of intraplasmic Ca^2+^. This triggers an intracellular signaling cascade that promotes the binding of talin to the intracellular end of the b2-chain, which leads to the activation of integrin. The extracellular pathway of α_IIb_β_3_ integrin activation may be caused by ligand binding (e.g., fibrinogen). The activation of α_IIb_β_3_ leads to the availability of the receptor for ligands that can bridge to other α_IIb_β_3_ on adjacent platelets.

The last group of developed and clinically tested antiplatelet agents were GPIIb-IIIa blockers (abcximab, eptifibatide and tirofiban). It should be noted that glycoprotein IIb-IIIa (integrin α_IIb_β_3_, CD41/CD61) still remains a promising target for antiplatelet therapy, which is also due to the extensive clinical experience of using this group of drugs in various categories of patients. This rationale became the foundation of the new antithrombotic strategy, α_IIb_β_3_ antagonism. GPIIb-IIIa inhibitors were found to be effective, and the observed adverse effects and complications contributed to a better understanding of the role of the GPIIb-IIIa adhesive aggregation function of platelets and opened up prospects for the development of other approaches to the inhibition of GPIIb-IIIa [11]. However, the long-term therapeutic utility of these agents is limited by the lack of their oral activity and their short duration of action. The development of oral forms of GPIIb-IIIa blockers is a promising direction for the development of antiplatelet therapy. One of the last well-studied oral GPIIb-IIIa blockers is orbofiban, an ester-type prodrug that forms active free acid in the liver, which is an orally active GPIIb-IIIa antagonist [12]. We identified a potential target for compound **1** and compared it to the indicators of the antiaggregant activity of orbofiban and acetylsalicylic acid, as the most selective antiaggregant agent. Compound **1** was chosen as the leader compound due to the available relatively simple and well-reproducible synthesis method with a high yield of the target product [13].

## 2. Materials and Methods

### 2.1. Synthesis of Compound ***1***

To a solution of isobornanthiol (2.21 g, 13 mmol) in ethanol (30 mL), Cs_2_CO_3_ (4.24 g, 13 mmol) and tetrabutylammonium iodide (4.80 g, 13 mmol) were added in an argon atmosphere. After 5 min of stirring, 2-bromoethanol (710 mL, 10 mmol) was added dropwise, and then the reaction mixture was refluxed for 24 h. After filtration, the clear solution was concentrated in vacuo, and the residue was purified via column chromatography on Acros silica gel with a 60–200 mesh using petroleum ether-EtOAc (1:1) as an eluent. Compound **1** was obtained as a liquid in a 90% yield, $[{\alpha ]}_{D}^{20}$ = +248.0° (0.3 c; CHCl_3_).

### 2.2. Molecular Docking

Terpenoid-integrin and orbofiban-integrin system modeling was carried out using the AutoDock 4.2 free software [14]. The geometry of molecules **1**, **2**, and **4**–**6**, and orbofiban was fully optimized using CAM-B3LYP/def2-TZVP [15,16] in the Gaussian 16, Revision C.01 program package [17]. The geometry of compound **3** was obtained using an X-ray [13].

The structure of the receptor was taken from a protein database (https://www.rcsb.org (accessed on 22 November 2023), ID = 3fcu). Small molecules were removed from the receptor’s structure, but metal ions were not removed, because they are involved in the binding of ligands. Additional optimization of the receptor structure was not performed. Due to the size of the receptor, docking was carried out in two separate calculations, in each of which the grid box was selected in such a way that, as a result of two calculations, all possible terpenoid binding sites were taken into account. For one calculation, the size of the grid box was 126 Å × 100 Å × 80 Å with a step of 0.750 Å, and for two calculations, it was 126 Å × 126 Å × 110 Å with a step of 0.700 Å. After determining and calculating the grid, molecular docking was carried out using the Lamarck genetic algorithm (LGA) [18]. Each calculation consisted of 50 separate runs, which ended after reaching 25,000,000 energy calculations. From the results obtained, the system with the lowest binding energy was chosen as the most stable. Molecular plots were made via UCSF Chimera [19].

### 2.3. Investigation of Antiaggregant Activity In Vivo

Experimental work in vivo was performed on 110 white non-linear male rats weighing between 250 and 300g in accordance with the recommendations of the “Handbook for preclinical study of new pharmacological substances” [20]. The animals were kept in standard vivarium conditions under natural lighting conditions, with an air temperature of 20 ± 2 °C and humidity at 55–60% in plastic cages with a litter of sawdust. Briefly, 24 h before the start of the research, feeding was stopped without restricting access to water. All experiments were carried out in compliance with the International Recommendations of the European Convention for the Protection of Vertebrates for Experimental Animals, the rules of laboratory practice to be followed during preclinical studies in the Russian Federation (GOST 3 51000.3-96 and 51000.4-96, GOSTR 50258-92) and the order of the Ministry of Health and Social Development of Russia, No. 708n, dated 23 August 2010, “On approval of the Rules of Laboratory Practice” (GLP).

The amount of the injected substance was calculated using the volume of the injected solution depending on body mass, taking into account the maximum allowable amount of fluid during intragastric administration. Acetylsalicylic acid at a dose of 20 mg/kg (an equimolar dose of IC50 acetylsalicylic acid obtained in vitro experiments) was used as a comparison drug [8]. Compound **1** and comparison drugs were administered orally 2 h before the study in equimolar doses of Acetylsalicylic acid. For compound **1**, this dose was 40 mg/kg. For the purpose of further calculations of ED50, the doses of substances were reduced or increased depending on their antiplatelet effect. For substance **1**, the doses used in the experiment were 10, 25, 35 and 40 mg/kg, while for acetylsalicylic acid and orbofiban, they were 20, 30 and 40 mg/kg. Before taking the material for the study, the animals were anesthetized with sodium thiopental (40 mg/kg intraperitoneally). Blood samples were stabilized with 3.8% sodium citrate solution in a ratio of 9:1. The blood was taken from the abdominal aorta of rats 1.5 h after the introduction of the compounds, then subjected to centrifugation for 10 min at 1500 rpm on a centrifuge OPN-3.02 (TNK “DASTAN”, Bishkek, Kyrgyzstan) to obtain PRP. Each sample contained 850–950 thousand/mL of platelets. The study of the effect of substances on the functional activity of platelets was carried out according to the Born G. [21] method on the aggregometer “AT-02” (NPF “Medtech”, Saint Petersburg, Russia) on PRP. ADP at a final concentration of 20 µg/mL was used as an inducer of platelet aggregation.

### 2.4. FACs Analysis

The experiments were performed in vitro on the blood samples of healthy male donors aged 18–24 years. The total number of donors was 12. Blood sampling for the study of compounds in relation to the hemostasis system was carried out from the cubital vein using BD Vacutainer^®^ vacuum blood sampling systems (Becton Dickinson and Company, Jersey, NJ, USA). A 3.8% sodium citrate solution in a ratio of 9:1 was used as a stabilizer of venous blood. Platelet-rich plasma samples were obtained via the centrifugation of citrated blood at 1000 rpm for 10 min, repeating the procedure for platelet-free plasma at 3000 rpm for 10 min. The centrifuge OPN-3.02 (TNK “DASTAN”, Kyrgyzstan) was used during the experiment. Cytofluorimetric analysis was performed on BD FACS Canto II (Becton Dickinson Immunocytometry Systems, San Jose, CA, USA) using original software. The expression of P-selectin on the platelet surface was used as a marker of platelet activation. The binding of fluorescently labeled antibodies against CD62 to blood platelets of healthy donors was measured. In order to carry this out, platelet-rich plasma samples were diluted 100 times with a 0.15 M phosphate salt buffer solution (pH 7.0–7.5), and the studied preparations were incubated for 5 min. To activate the platelets, ADP was introduced into the samples to reach a final concentration of 20 micrograms/mL and mixed thoroughly. Activation was carried out for 15 min, after which the cells were fixed by adding 1% Formalin solution. After incubation, platelet-rich plasma samples were stained for 20 min at room temperature with mouse anti-CD62 monoclonal antibodies (mAbs) labeled with APC (Alophycocyanin) (Becton Dickinson, USA) in accordance with the manufacturer’s recommendations. To assess the binding to the GPIIb-IIIa receptor, we examined the expression of CD41a and CD61 on the platelets by using the corresponding mAbs, conjugated with (phycoerythrin) and FITC (fluorescein isothiocyanate), respectively (Becton Dickinson, USA). The same instrument settings were used for all measurements. At least 10,000 events were counted for each sample. The “platelet window” was distinguished via the parameters of direct (FCS) and small-angle (SSC) light scattering in the logarithmic coordinate scale. The number of positive cells (%) was estimated via the expression of CD62. Acetylsalicylic acid was used as a negative control, and the GP blocker IIb–IIIa, tirofiban, at a concentration of 10^−3^ M was also used as a negative control.

Comparison drugs:− Orbofiban (*N*-[[(3*S*)-1-(*p*-amidmophenyl)-2-oxo-3-pyrrolidinyl]carbamoyl]-β-alanine, ethyl ester; Skokie, IL, USA);− Acetylsalicylic acid (2-acetylbenzoic acid; Pharmaceutical plant Shandong Xinhua Pharmaceutical Co, Ltd., Zibo, China).− Tirofiban ((2*S*)-2-(butylsulfonylamino)-3-[4-(4-piperidin-4-ylbutoxy)phenyl]-propanoic acid; hydrochloride; Siegfried Hameln, Hameln, Germany).

Statistics:

The results were processed using the Statistica 10.0 statistical package (StatSoft Inc., Tulsa, OK, USA). Tests for the normality of the distribution of actual data were carried out using the Shapiro–Wilk criterion. The median and interquartile interval were used to describe the groups. The significance level of p for the statistical criterion was set at 0.05. The ED50 values were calculated using the Prism program (GraphPad Software, Inc., La Joll, CA, USA) via the nonlinear fitting of a curve describing activity (%) in a logarithmic equation with four parameters.

### 2.5. Cells Viability MTS Assay

BJ tert fibroblasts were seeded into the 96-well flat-bottomed plates (Corning Inc., Corning, NY, USA) and cultured for 24 h. The cells were further treated with tested compounds at concentrations indicated below or with DMSO as a negative control for 72 h. To assess cellular viability, MTS reagent (Promega, Madison, WI, USA) was introduced into the cell culture, and incubation was followed for 1 h. The MTS reduction product was measured using a MultiScan FC plate reader (Thermo Fisher Scientific, Waltham, MA, USA) at 492 nm. IC50 was calculated as the concentration of the compound inhibiting cell growth by 50% after 48–72 h. Data were normalized to the DMSO-treated (control) group. IC50 values were determined by using IC50 Tool Kit (http://ic50.tk/, accessed on 10 January 2023).

## 3. Results and Discussion

### 3.1. Molecular Docking

We evaluated the affinity of thioterpenoids **1**–**6** to integrin α_IIb_β_3_ using molecular docking. Compound **2** is characterized by the presence of two stable conformers (**2c1** and **2c2**), which differ in the OH group rotation angle. Compound **3** is a dimer, so molecular docking was carried out for each of the monomers (**3m1** and **3m2**) in the dimer.

The models obtained using molecular docking made it possible to theoretically determine the most probable binding site and the binding affinity of terpenoids (Figure 2A) and orbofiban (Figure 2B) to integrin.

Of all the terpenoids, two (**3m1** and **6**) were found to be localized at the same binding site within integrin, which also includes magnesium and calcium ions. The amino acid composition of this binding site was as follows: SER121, TYR122, SER123, PHE160, TYR166, TYR190, ARG214, ASN215, ARG216, ALA218, GLU220, ALA252, Mg2001, Ca2003 for **3m1** (Figure 3) and SER121, TYR122, SER123, PHE160, TYR166, MET180, TYR190, SER213, ARG214, ASN215, ARG216, ASP217, ALA218, GLU220, Mg2001 and Ca2003 for **6** (Figure 3). The obtained thermodynamic parameters of docking allow us to conclude that the efficiency of terpenoid binding is similar. The binding energy was −5.73 kcal/mol for molecule **3m1** and −5.86 kcal/mol for molecule **6**, mainly due to the sum of van der Waals interactions, hydrogen bonding and desolvation energy. Terpenoids are actively involved in the formation of hydrogen bonds due to carbonyl and hydroxyl oxygen. The remaining terpenoids bind within integrin at other binding sites with similar binding energies. The composition of binding sites for the remaining terpenoids was as follows: TYR189, LEU192, GLY193, LEU194, SER222, PHE223, ASP224, SER225, TYR230, TRP235, GLU268, LEU270, TYR274, GLN275, ARG276 for molecules **1**, **2c1** and **2c2** (Figure 3), LYS189, LEU192, LEU194, GLY198, SER222, PHE223, ASP224, SER225, TYR230, TRP235, GLU268, LEU270, TYR274, GLN275, ARG276 for molecule **3m2** (Figure 3), PRO40, ARG41, GLU49, ALA89, ARG90, GLN91, GLY92, TRP110, GLN111, HIS112, PRO126, VAL161, SER162, PRO163, LEU262, ALA263, GLY264 for molecule **4** (Figure 3) and PRO268, ASN269, ASP270, GLY271, ASP288, TYR289, PRO290, SER291, LEU294, ALA257, LEU312, GLU324, TYR353, ARG355 and TYR380 for molecule **5** (Figure 3).

In addition to the molecular docking performed for **1**–**6** with integrin α_IIb_β_3_, we carried out molecular docking for the control compound, orbofiban, with integrin α_IIb_β_3_. The docking results reveal that, unlike compounds **1**–**6**, which exhibit affinity for the propeller domain of integrin α_IIb_β_3_, orbofiban is localized near the α-helix of the βA domain of integrin α_IIb_β_3_. The presence of a substantial number of centers for intermolecular interactions (van der Waals, H-bonding, and π-stacking) in the orbofiban molecule with reactive amino acid residues (mainly LYS321, LEU322, LEU352 and TYR353) of the integrin α_IIb_β_3_ determines its comparatively high affinity for the receptor, with a binding free energy value of −5.72 kcal/mol (Table 1). The complete amino acid composition of the binding site for orbofiban in integrin includes the following amino acid residues: GLY293, LEU294, THR296, GLU297, LEU299, SER300, ILE304, ASN305, LEU306, ARG320, LYS321, LEU322, ALA323, GLU324, ILE325, PRO326, GLY327, THR328, LEU352 and TYR353 (Figure 3). The localization of orbofiban in integrin, in a domain different from that of compounds **1**–**6**, as well as its distance from metal ion-dependent adhesion sites (MIDAS) and ligand-associated metal binding sites (LIMBS or SyMBS), indicates distinct mechanisms of their antithrombotic activity. Moreover, the comparable affinities of orbofiban and compounds **1**–**6** for integrin α_IIb_β_3_ suggest the formation of stable supramolecular complexes between these small ligands and the receptor.

The binding energies of these terpenoids differ slightly from the values for molecules **3m1** and **6**; **2c2** binds most effectively (Table 1).

Such binding energy values indicate the formation of stable supramolecular complexes of thioterpenoids **1**–**6** with integrin. The presence of calcium and magnesium cations in the binding site indicates that the binding of molecules to the receptor can have a significant effect on the interaction, for example, with fibrinogen, as well as on the activity of the receptor as a whole, but this issue requires more thorough experimental research. Taking into account the similar binding parameters of all compounds, compound **1** was chosen for subsequent experiments due to the available relatively simple and well-reproducible synthesis method with a high yield of the target product.

### 3.2. Determination of Antiaggregant Activity during Intragastric Administration to Rats

The experiments made it possible to establish the presence of a pronounced dose-dependent antiplatelet effect in compound **1** when administered intragastrically to rats. Compound **1** at a dose of 40 mg/kg inhibited platelet aggregation by 71.4%. When the dose was reduced to 30 mg/kg, the percentage of inhibition corresponded to 62.5. As a result of further dose reduction to 20 mg/kg, the amplitude of platelet aggregation increased, and the calculated value of ED_50_ was 35.1 mg/kg (Table 2).

The comparison drug, acetylsalicylic acid, also dose-dependently inhibited the functional activity of rat platelets. At doses of 20, 30 and 40 mg/kg, this substance reduced platelet aggregation by 32.7, 54.8 and 63.2%, respectively. At the same time, the ED50 of acetylsalicylic acid was 23.7 mg/kg. For orbofiban, this dose was 22.9 mg/kg.

Thus, compound **1** exhibits antiaggregant activity when administered intragastrically to rats at levels relatively close to those of acetylsalicylic acid and orbofiban.

### 3.3. FACs Analysis

The results illustrating the effects of the tested compounds on platelet activation and binding to the GPIIb-IIIA receptor are shown in Table 3.

It was found that acetylsalicylic acid does not affect the expression level of CD62. Compound **1**, similarly to the drug of the GPIIb-IIIa blocker group, tirofiban, effectively reduced the expression level of P-selectin to the values similar to those of the intact platelets. Tirofiban became bound to the CD41a integrin of the GPIIb-IIIa receptor, and the degree of binding depended on the concentration of the drug. Compound **1** demonstrated direct binding to platelet GPIIb-IIIa receptor CD61 integrins at the studied dosages.

### 3.4. Cytotoxicity

The cytotoxic properties of compound **1** was evaluated in a metabolic MTS-based assay. For this, BJ tert human fibroblasts (ATCC, Manassas, VA, USA) were seeded in 96-well culture plates (Corning Inc. Corning, NY, USA) and allowed to attach and grow for 24 h before treatment. Then, compound **1** was introduced into cell culture and cells were cultured with compound 1 for 72h to assess cellular viability. We found that compound **1** exhibited the cytotoxic properties against BJ tert fibroblasts at concentrations higher than 300 µM (Figure 4), with IC50 equaling 361+/−28 (7.8%) µM.

## 4. Conclusions

The results of molecular docking confirm the high affinity of terpenoids **1**–**6** for integrin, including through binding sites containing Mg^2+^ and Ca^2+^ ions. This may indicate their potential inhibition of platelet aggregation by reducing the affinity of integrins for fibrinogen. When assessing the antiaggregational effect of the intragastric administration of it to rats, thioterpenoid **1** showed activity at the level of that of acetylsalicylic acid and the oral GP IIb-IIIa blocker orbofiban. Compound **1**, similarly to the drug of the GPIIb-IIIa blocker group, tirofiban, effectively reduced the expression level of P-selectin to the values similar to those of the intact platelets. Tirofiban became bound to the CD41a integrin of the GPIIb-IIIa receptor, and the degree of binding depended on the concentration of the drug. Compound **1** demonstrated direct binding to platelet GPIIb-IIIa receptor CD61 integrins at the studied dosages.

## Figures and Tables

**Figure 1 biomimetics-08-00570-f001:**
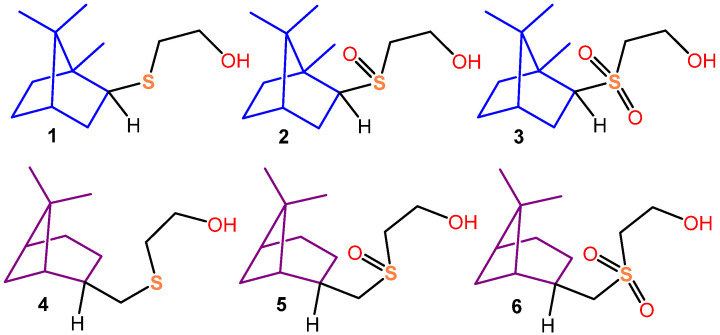
Structures of compounds **1**–**6**.

**Figure 2 biomimetics-08-00570-f002:**
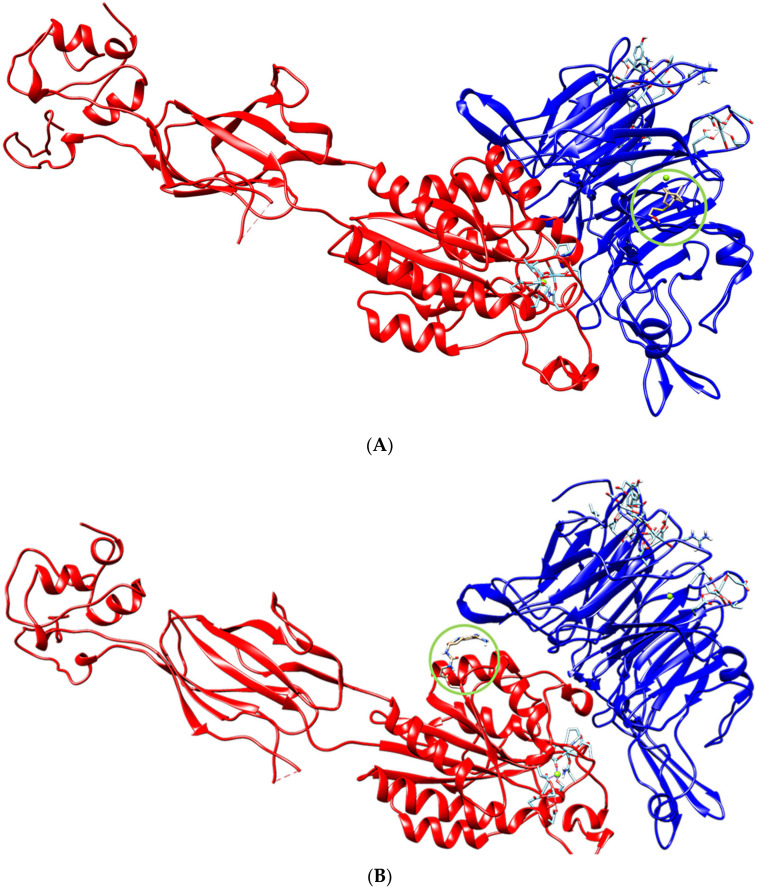
The localization of molecule **1** (**A**) and orbofiban (**B**) in integrin.

**Figure 3 biomimetics-08-00570-f003:**
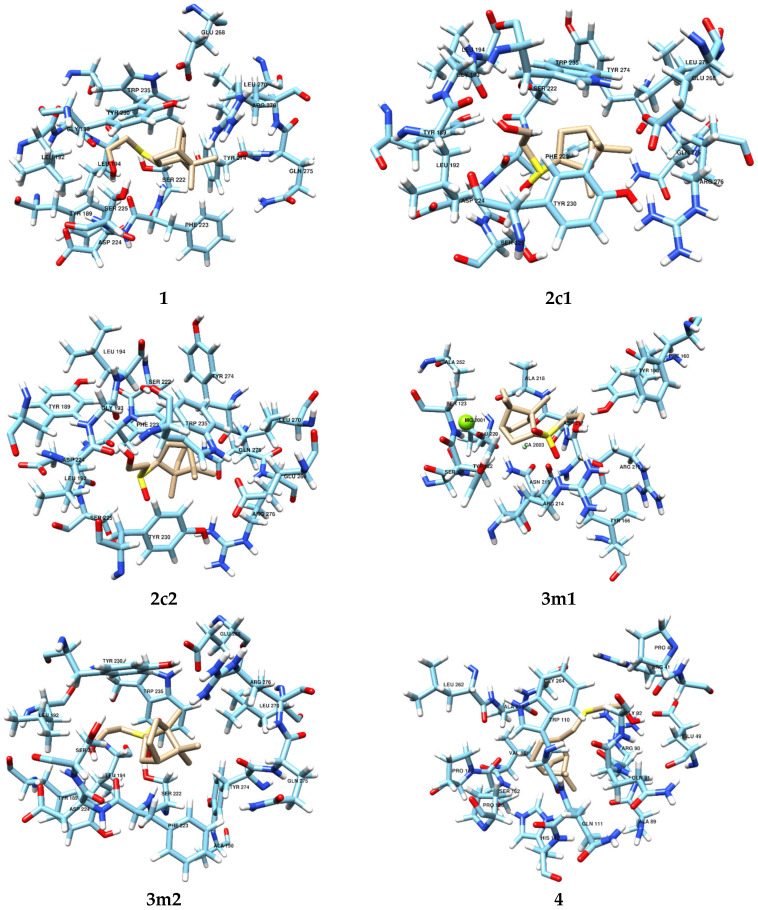
Terpenoid and orbofiban localization at the most likely binding site (the composition of the integrin–terpenoid and orbofiban–integrin binding site).

**Figure 4 biomimetics-08-00570-f004:**
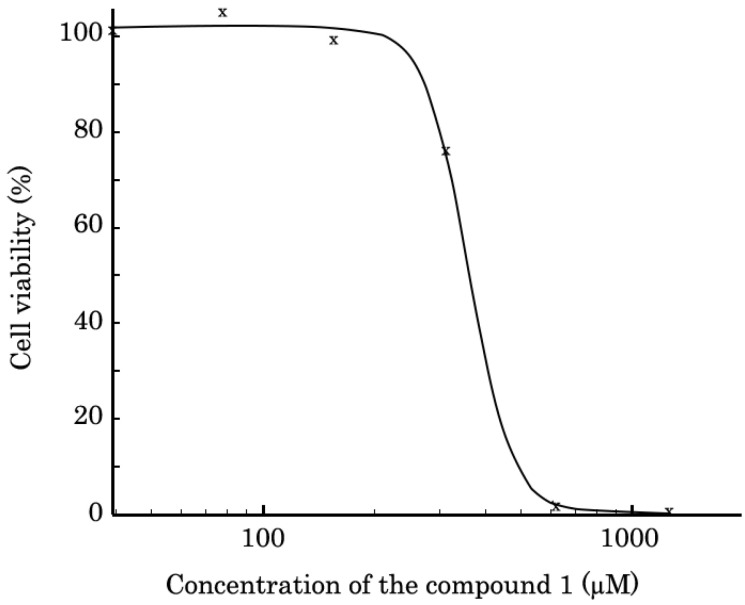
The viability of BJ tert human fibroblasts treated compound 1 for 72 h. The data was normalized to solvent (e.g., DMSO)-treated controls. Values are the means (x) ± standard deviation (*n* = 3). Statistical analyses (Student’s *t*-test) were performed using Statistical software program version 7.0 (S.A. Glantz, McGraw Hill Education, New York, NY, USA). *p* < 0.05 was considered to indicate a statistically significant difference. Half-inhibitory concentration (IC50) was determined by using the IC50 Tool Kit (http://ic50.tk/, accessed on 10 January 2023). The IC50 value is graphically represented as a proper sigmoid curve and equals 361+/−28 (7.8%) µM.

**Table 1 biomimetics-08-00570-t001:** Binding energy values for terpenoid–integrin and orbofiban–integrin complexes.

Compound	Binding Energy, kcal/mol
**1**	−5.67
**2c1**	−5.83
**2c2**	−5.96
**3m1**	−5.73
**3m2**	−5.68
**4**	−5.44
**5**	−5.65
**6**	−5.86
**orbofiban**	−5.72

**Table 2 biomimetics-08-00570-t002:** Indicators of ED50 of compound **1** and comparison drugs on the model of the ADP-induced aggregation of rat plasma platelets with intragastric administration; Me (0.25–0.75), *n* = 10.

Compound	Dose, mg/kg	% Inhibition of Platelet Aggregation	ED_50_, mg/kg
**1**	40	71.4 (68.5–73.2) *	25.9
	30	62.5 (60.1–64.8) *	
	20	35.1 (32.4–38.3) *	
Acetylsalicylic acid	40	63.2 (57.4–65.2) *	23.7 ^‡^
	30	54.8 (50.1–56.9)*	
	20	15.2 (14.8–17.3) *	
Orbofiban	40	80.4 (76.5–83.1) *	22.9 ^‡^
	30	52.7 (49.4–53.6) *	
	20	18.3 (16.5–21.9) *	

Note: *—the data are reliable with respect to the control at *p* < 0.05; ^‡^—acetylsalicylic acid /orbofiban vs. compound **1** at *p* < 0.05.

**Table 3 biomimetics-08-00570-t003:** Indicators of platelet activation in the presence of compounds and comparison drugs; Me (0.25–0.75), *n* = 10.

Compound	CD62		GP IIb-IIIa, IC50	
	ADP−	ADP+	CD41a	CD61
**1**	1.1 (0.9–1.2)	1.2 (1.1–1.5) ^‡^	n/e	1.2 × 10^−2^
Tirofiban	1.2 (1.1–1.4)	3.5 (2.7–4.1) ^‡‡^	1.9 × 10^−3^	n/e
Acetylsalicylic acid	1.3 (1.1–1.4)	16.4 (14.5–17.3) ^‡‡^	n/e	n/e

Note: the level of statistical significance of the differences in the characteristics of the groups after ADP activation: ^‡^—*p* > 0.05; ^‡‡^—*p* ≤ 0.05. CD62ADP−—CD62 expression before ADP exposure; CD62ADP+—CD62 expression after ADP exposure. n/e—no effect.

## Data Availability

Data are contained within the article.

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
