# Peer review of "Unraveling the Mechanism of Platelet Aggregation Suppression by Thioterpenoids: Molecular Docking and In Vivo Antiaggregant Activity"

_biomimetics, 2023, doi:10.3390/biomimetics8080570_

Round 1
Reviewer 1 Report
Comments and Suggestions for Authors
The present article by Nikitina and co-workers describes the study of the effects of a collection of thiomonoterpenoids on platelet aggregation using molecular docking techniques. The authors claimed that all compounds were active inhibitors of platelet aggregation. In addition, the authors selected compound 1 for further studies on its antiaggregant activity in rats. The authors described that the effects were similar to those of other existing drugs, such as acetylsalicylic acid and orbofiban.
Generally, this is a good work that provides important information about alternative compounds for the inhibition of platelet aggregation, providing new pharmacological tools for the treatment of disorders in blood coagulation. Therefore, I recommend accepting this article under MINOR revision.
Some considerations must be considered.
1.- Along the manuscript, the authors use the names “acerylsalicylic acid”, “orbofiban” and “tirofiban” in capital letters in any part of the text. This is not correct; the authors must change to the proper form in any part of the text.
2.- Page 7, Figure 2. The position of compound 1 is not visible in the figure. The author could include a narrow to facilitate the visibility of the compound or focus on the figure in the zone where 1 appears.
3.- Page 9, Table 2. The authors describe the results of ED50 for compound 1 compared with those for acetylsalicylic acid and obofiban. However, the doses employed in the study were different for 1 compared with the other compounds. I do not understand this fact. The authors have to justify in the manuscript why they used similar (but different) doses of compound 1.
4.- Many typing mistakes can be detected in the text. A full revision is necessary.
Comments on the Quality of English LanguageMinor editing of English language required
Reviewer 2 Report
Comments and Suggestions for Authors
I am delighted to have the opportunity to review the manuscript. In general, the manuscript is well-crafted; however, I have a few minor suggestions that I believe could enhance its strength:
1. The experiments have been described and demonstrated quite effectively. I recommend that the author consider incorporating molecular dynamics, which would bolster the insilico study.
2. In the context of molecular docking, I highly recommend that the author perform molecular docking with the market drug compound, which can serve as a control.
3. In the discussion section, I propose that the author provide a more detailed description of the results.
After these corrections, the article will be ready for acceptance.
Thank you
Reviewer 3 Report
Comments and Suggestions for Authors
Dear Authors
Thank you for giving me the chance to look at your work, which was quiet interesting. I have some some comments listed below,
Title: Suggest the change of title to : Molecular Docking and In vivo Antiaggregant Activity
At all parts including abstract clearly outline that only compound 1 is synthesized and used for invivo experiment.
Line |
Comment |
Abstract |
|
36 |
Bicyclic monoterpenoids with two different skeleton |
39 |
Compound 1 a pinane….. |
|
The enantiomer if compound 1 |
122 |
In vitro italics throughout text |
171 |
Silica gel source |
173 |
0.3 is Rf what system |
Comments on the Quality of English Language
English editing and grammar check recommended. also for better text flow and connectivity
